# Microstructure and Texture Characterization of Duplex Stainless Steel Joints Welded by Alternating Magnetic Field-Assisted Hybrid Laser-GMAW Welding

**DOI:** 10.3390/ma15248741

**Published:** 2022-12-07

**Authors:** Juan Fu, Zhipeng Rao, Yong Zhao, Jiasheng Zou, Xin Liu, Yanfei Pan

**Affiliations:** 1Provincial Key Lab of Advanced Welding Technology, Jiangsu University of Science and Technology, No. 2 Mengxi Road, Zhenjiang 212003, China; 2Jiangsu Changjiang Intelligent Manufacturing Research Institute Co., Ltd., Changzhou 213000, China

**Keywords:** alternating magnetic field, hybrid laser-GMAW welding, 2205 duplex stainless steel, microstructure, texture characterization

## Abstract

In this study, 2205 duplex stainless steel with 12 mm thickness was welded by alternating magnetic field-assisted laser arc hybrid welding. The effect of an alternating magnetic field on the proportion distribution of two phases of the welded joint was investigated. The texture distribution, grain boundary misorientation, and grain size of welded joints were analyzed and characterized. The uniform distribution of alloying elements in the two phases was improved by a 20 mT alternating magnetic field. The diffusion dissolution of Ni and N elements into the γ phase was promoted, which was conducive to the transition from the α to γ phase and reduced the precipitation of Cr_2_N, such that the ratio of γ to α was 43.4:56.6. The ratio of the two phases of the weld was balanced by the alternating magnetic field of 30 mT, such that the ratio of γ and α was 44.6:55.4 and the texture dispersion was weakened. The Σ3 twinning boundary of the austenite phase in the weld was transformed to HABs, the ferrite phase underwent dynamic recrystallization, and the austenite phase had a cube texture, copper texture, and goss texture.

## 1. Introduction

The balanced duplex ratio in the microstructure of 2205 duplex stainless steel determines its superior yield strength, chloride stress corrosion resistance, and welding ability that can be used in the shipbuilding and pressure vessel industry [1,2]. After the complex welding thermal cycle, the microstructure distribution of the 2205 duplex stainless steel welded joint is the key factor affecting the performance of the whole welding [3,4,5]. In order to ensure the excellent performance of 2205 duplex stainless steel welded joints, the welding heat input and post-welding cooling rate should be controlled, and the welding efficiency should be considered when welding thick plates. Hybrid Laser-GMAW welding is an efficient and precise technology that combines two heat sources and has advantages in reducing residual welding stress and improving welding efficiency [6,7,8,9]. Unstable arc and difficult-to-control droplet behavior are the main factors causing defects, such as the non-fusion of the sidewall in the narrow-gap hybrid Laser-GMAW welding of medium and thick plates [10,11,12,13]. Therefore, how to regulate the welding process reasonably or additional auxiliary measures to improve the stability of arc and droplet is the focus of studies worldwide.

Magnetically controlled arc technology is an advanced welding technology that applies an electromagnetic field to the welding process to adjust arc morphology, control droplet behavior, and improve the microstructure and properties of the joints [14,15]. When the magnetic field is applied to the hybrid Laser-GMAW welding, the shape and motion mode of the arc plasma and photoinduced plasma are altered under the action of Lorentz force, which in turn controls the stability of the welding process and improves the energy utilization rate. On the other hand, the flow state of the molten pool is regulated by an electromagnetic stirring principle to improve the distribution of elements and refine the grains [16,17]. Presently, the effects of various magnetic field types on welding phenomena (arc morphology and droplet transition) and grain refinement have been studied [18,19]. However, the effect of an alternating magnetic field on the microstructure transformation of a hybrid Laser-GMAW welding joint of duplex stainless steel has not yet been investigated. The phase distribution of duplex stainless steel welding joint organization in improving the workpiece plays a critical role, but the alternating magnetic field on 2205 duplex stainless steel hybrid Laser-GMAW welding of the phase transition mechanism is not clear; therefore, studying the microstructure and properties of duplex stainless steel joints welded by alternating magnetic field assisted hybrid Laser-GMAW welding is significant.

In this study, the welding process and microstructure of duplex stainless steel weld prepared by alternating magnetic field-assisted hybrid Laser-GMAW welding were studied [20,21,22]. Optical microscopy (OM), scanning electron microscopy (SEM)-energy dispersive spectroscopy (EDS), X-ray diffraction (XRD), and electron backscatter diffraction (EBSD) [23] were used to analyze and characterize the microstructure evolution and reveal the influence of alternating magnetic field on the element distribution and microstructure of welded joints.

## 2. Experimental Procedures

### 2.1. Materials and Welding Process

The base material used in this experiment was hot-rolled 2205 duplex stainless steel sheet with a 12-mm thickness. The two-phase microstructure of 2205 duplex stainless steel is shown in Figure 1, in which the base was ferrite (α phase), and the white was austenite (γ phase). Image-Pro statistical results showed that the contents of austenite and ferrite in the base material accounted for 43.83% and 56.17%, respectively. The filler wire was ER2209 stainless steel solid wire with a 12-mm diameter. The chemical composition of DSS2205 and ER2209 is shown in Table 1. The alternating magnetic field-assisted hybrid Laser-GMAW welding was used to prepare duplex stainless steel joints. The 12-mm thick duplex stainless steel had a 4-mm blunt edge on one side, the groove angle was 6°, and the welding torch angle was about 60°. Pure laser spot welding was used to position the plate before welding to prevent deformation. The experimental process is illustrated in Figure 2. A YLS-10000-SS4 fiber laser generator (wavelength 1025–1080 nm), a TPS500i intelligent arc welding power system, and a KR60HA six-axis industrial robot were used in the test, which could accurately control the welding trajectory and maintain a stable welding speed. The alternating magnetic field equipment was a Jetline Engineering 8080 magnetic arc control system equipped with a 4604 head, the alternating magnetic field frequency range was 0–50 Hz, and the magnetic induction intensity range was 0–30 mT. The welding method without a magnetic field was used to remove the magnetic coil, and the other processes were the same as welding with a magnetic field.

### 2.2. Microstructural Characterization

The metallographic samples in vertical weld direction were prepared. Zeiss optical microscopy (OM) was used to observe each area of the welded joint, and Image Pro 6.0 software was used to calculate the ratio of the α/γ phase. Regulus-8100 scanning electron microscope (SEM) and energy dispersive spectrometer (EDS) were used to observe the precipitate and determine the distribution of elements in each region of the specimen. Electron backscatter diffraction (EBSD) analysis was used to explore the influence of alternating magnetic fields on the grain orientation characteristics of the two phases. Channel 5 software was used to analyze the texture and grain orientation of the samples. A Brooke D8 Advance X-ray diffractometer was used to scan the weld surface and base material from 10 to 90° with a time step of 0.15 s and a 2θ scan step of 0.02° through a Cu target Cu-Ka radiation. Finally, the phase was analyzed, and the crystal structure was determined using Jade 6.5 analysis software.

## 3. Results and Discussion

### 3.1. Effect of a Transverse Alternating Magnetic Field on the Weld Microstructure

#### 3.1.1. Effect of an Alternating Magnetic Field on the Microstructure Characteristics of a Welded Joint of 2205 Duplex Stainless Steel

As shown in Figure 3, with the increase in the alternating magnetic field, the ferrite grains in the weld became small, and the grain boundary austenite grew into the ferrite and crossed the whole ferrite grains. The austenite had a distinct columnar structure and abundant dendritic crystals. The physical characteristics of 2205 duplex stainless steel and alternating magnetic field changed the heat distribution in the molten pool, resulting in prolonged high-temperature residence time and a growing tendency of bulk γ grains. Table 2 shows the γ/α phase ratio obtained after calculation and statistics in the weld (WM). The content of γ phase was lower than α phase. When the B was 30 Mt, the ratio of γ phase and α phase reached 44.6:55.4, which was beneficial to the mechanical properties and corrosion resistance of the welded joint.

#### 3.1.2. Study on the Distribution of Welded Joint Elements

Figure 4 shows the element line scanning results of the two-phase microstructure in the 20-mT alternating magnetic field weld. Based on the variation trend of the element curve, the distribution of Fe, Cr, and Ni in the γ phase and α phase had few differences, which was mainly due to the welding which is a rapid heating and cooling process, and the elements do not have sufficient time to diffuse. However, the distribution curve of Mo exhibited a decreasing trend, showing a high and dense peak in α phase, indicating that the distribution of Mo was different between the two phases. This is because Mo is an element that strongly forms and stabilizes the α phase and shrinks the γ phase region. In the process of weld cooling crystallization, the γ phase is mainly formed in the solid phase transformation of α phase. While α phase is constantly consumed, residual Mo in the α phase is concentrated, eventually increasing the content of Mo in α phase than in γ phase. The results showed that Mo improved the pitting resistance of 2205 duplex stainless steel and promoted the precipitation of the phase, such that the content of Mo should match the content of Ni and N to maintain the uniformity of the two phases. 

SEM and EDS technology were used to analyze the distribution differences of Cr, Ni, Mo, and N in austenite and ferrite in the weld under an alternating magnetic field, as shown in Figure 5. Table 3 shows the test results of two-phase EDS at different locations. The graph shows a marked difference in the element content between BM and WM; Cr and Mo were more in the α phase, and Ni and N in γ phase were higher than in the α phase, indicating that element diffusion is a major factor affecting tissue phase transition. Compared to the base metal, the content of Cr in the ferrite phase in the weld decreased, mainly because the content of the α phase in the weld increased, and the original α phase forming element Cr in the weld was diluted with the increased α phase volume. Another α phase forming element of Mo did not decrease but increased with the addition of welding wire elements. The main changes were related to the fact that with the application of an alternating magnetic field, the content of Mo in ferrite decreased from 6.16% to 5.87%, while the content of Ni in austenite increased from 6.91% to 7.32%, indicating that the alternating magnetic field affected the diffusion of elements in the weld. The alternating magnetic field at a certain frequency changed the plasma morphology, reduced the droplet transition frequency, and increased the droplet temperature. Simultaneously, the oscillation of the molten pool improved the heat distribution, such that the Ni in the α phase could fully spread to the γ phase and promote the transition from α to γ phase. In addition, the element content difference between the two phases of the weld after the application of alternating magnetic field was smaller than that of the weld without the magnetic field, indicating that the alloying element diffusion under the action of the alternating magnetic field was uniform and the element segregation was avoided. When the content of the γ phase increased, N diffused and transformed from the α to γ phase, so that the N content in austenite was higher than that in the ferrite phase. This in turn, avoided the supersaturated precipitation of N in α phase, forming Cr_2_N.

#### 3.1.3. XRD Analysis of Weld

Figure 6 shows the XRD patterns of the BM, the WM without a magnetic field, and the WM with a 20 mT alternating magnetic field. The samples were composed of γ and α phases, and no secondary phases (σ phase, Cr_2_N) and inclusions were observed. Most of the γ and α phases precipitated along the most close-packed plane: (111) γ and (110) α. However, the optimal arrangements of the ferrite and austenite in the base metal were disrupted by welding heat sources and magnetic fields. In addition to (110) α close-packed plane, ferrite phases were formed in the (200) α and (211) α planes, and the austenite phases were arranged in (200) γ and (220) γ planes in addition to (111) γ plane in the WM without magnetic field; whereas, in the WM with the alternating magnetic field, the ferrite phase only formed in the (211) α plane except for the (110) α close-packed plane, and the austenitic phase formed diffraction peaks with low intensity in the (200) γ and (220) γ planes except for the (111) γ close-packed plane. This may be because the alternating magnetic field alters the texture distribution and grain size inside the WM, and the grain growth of the close-packed plane is hindered.

### 3.2. Effect of a Transverse Alternating Magnetic Field on the Grain Orientation and Texture of the Weld Microstructure

#### 3.2.1. Two-Phase Orientation and Texture Distribution of Welds

Figure 7 shows the EBSD phase diagram, IPFs in the RD direction, and band contrast (BC) of the weld under alternating magnetic field intensities. The distribution and content of the two phases in the structure can be calculated using the EBSD phase diagram. The phase diagram shows that after the alternating magnetic field was applied, the γ-phase in the weld structure increased and the grains of the α-phase became refined. The α phase precipitated a coarse γ phase at the grain boundary, and the intragranular austenite precipitated in the α phase had a tendency to grow up. The EBSD phase diagram shows that the application of an alternating magnetic field improved the two-phase proportional balance of weld microstructure. The IPF reflects the grain orientation, and the 0 mT IPF reflects the columnar grains of the non-magnetic field weld, which was mainly composed of coarse ferrite, fine grain boundary austenite precipitated from its grain boundaries, and fine fragmented intragranular austenite. The orientation of ferrite grains was random. The IPFs of the alternating magnetic fields of 20 and 30 mT showed that the columnar crystals grew from the edge to the center. The ferrite in the 20 mT alternating magnetic field had two preferred orientations of red and blue, and most of the ferrite grains were approximately parallel to the RD plane at the orientation of (111) plane and (001) plane. In the weld of 30 mT magnetic field, most of the ferrite grain (101) plane was parallel to the RD plane.

Figure 8 shows the austenite and ferrite pole figures of the welds under different alternating magnetic field intensities. The relative intensity of the texture can be reflected by the dark pole density on the pole figure. Compared to the standard (100), (110), and (111) projections of the cubic crystal system, the austenite grain orientation in the non-magnetic field WM was dispersing, and the maximum orientation distribution density was 5.7, while the maximum density of ferrite grain orientation was 13.89, indicating a strong preferred orientation of ferrite grains. The preferred orientation of grain increased with the increase in magnetic field intensity. When the alternating magnetic field density was 30 mT, the ferrite orientation distribution density was up to 19.01, and the austenite orientation distribution density was up to 6.59.

#### 3.2.2. Grain Boundary Orientation Difference and Grain Size of the Weld

Irregular arrangement of atoms and grain boundaries with high energy are major factors that affect the solid-state phase transition and properties of materials. According to the difference in orientation angle between adjacent grains, grain boundaries can be divided into low-angle grain boundaries (LABs) (orientation difference < 15°) and high-angle grain boundaries (HABs) (orientation difference > 15°), in which HAGBs include ordinary high-angle and coincidence site lattice (CSL) grain boundaries, and the Σ3 coincidence lattice of [111]/60° often occurs in austenite. The orientation correlation of the two CSLs is a twinning association, such that the CSL grain boundary content is identified as the Σ3 twin grain boundary content in this study.

Figure 9 shows the distribution of ferrite and austenite grain boundary orientation differences in welds under different alternating magnetic field intensities. Typically, the maximum orientation difference of a cubic crystal system was about 67°, and a deviation of about 1.5° was allowed in the measurement. Figure 9 shows that the austenite phase in the non-magnetic field weld contained about 55% of LABs and 18.7% of CSL grain boundaries, indicating an obvious bimodal distribution. In the ferrite phase, LABs were dominant, accounting for about 93% of the HABs that only accounted for about 6.7%, while the 0.3% twin boundaries were negligible. This was because the dislocation density in the ferrite phase was large, and it was easy to form a subgrain boundary structure. When an alternating magnetic field was applied during hybrid welding, dislocation climbing and grain boundary transition during grain formation were affected. When the alternating magnetic field strength was 20 mT, the proportion of LABs in austenite was significantly increased, and the Σ3 twin boundary was reduced to 9.7%, indicating that although the stacking fault energy of austenite was low, dislocation climbing was difficult to achieve; however, local dynamic recovery occurred under the action of the alternating magnetic field. When the alternating magnetic field strength was 30 mT, the HABs of the austenite phase increased by 3.9% compared to the non-magnetic field, and the Σ3 twin boundaries decreased by 7.6%, indicating that the effect of the alternating magnetic field promoted the rotation of the twin boundaries to shift to HABs. In addition, the LABs of the ferrite phase decreased to 89.9% as the alternating magnetic field strength increased to 30 mT, while the proportion of HABs increased. This phenomenon could be attributed to the high stacking fault energy of the ferrite phase, making dislocations an easy climb. The increase in the alternating magnetic field intensity changed the welding thermal cycle and caused dynamic recrystallization within the ferrite grains, resulting in a decreasing trend of LABs. These results indicated that the ratio of LABs and HABs affected the plasticity and toughness of the samples, and the existence of HABs can hinder the propagation of cracks.

### 3.3. Mechanism Analysis of the Effect of an Alternating Magnetic Field on the Tissue Structure

Figure 10 shows the effect of an alternating magnetic field on the growth and alignment of dendrites during solidification. G represents the temperature gradient, B represents the alternating magnetic field, current I is mainly the thermal current generated near the solid–liquid interface caused by Seebeck effect and the motional current generated by the liquid metal moving in the alternating magnetic field, FA represents the electromagnetic force generated by the interaction of the current and the alternating magnetic field, and FR Indicates viscous resistance. Figure 10a shows the growth of dendrites under the action of natural convection. The low-temperature solid-phase region represents the polygonal growth of ferrite in HAZ, and the solidification process was dominated by the growth of coarse ferrite columnar crystals. Figure 10b shows the growth of dendrites after applying an alternating magnetic field. The alternating magnetic field enhanced the heat transfer and convection of the melt, which reduced the temperature gradient and compositional supercooling of the solid–liquid front, thereby inhibiting the growth of columnar crystals. The electromagnetic force generated by the interaction of thermal and emotional currents with the alternating magnetic field drove the flow of the liquid phase in the molten pool, modifying the solute concentration between the dendrites, thus resulting in a solute concentration gradient. This changed the dendrite growth driving the force and growth rate, resulting in a tilted dendrite growth direction and additional branches. The electromagnetic stirring effect of the electromagnetic force on the molten pool caused the tip of the dendrite to break and remelt, increasing the amount of nucleation. Moreover, the arc plasma shrank, expanded, and oscillated under the alternating magnetic field, such that the molten pool was mechanically stirred, and the grain refinement effect was enhanced. Figure 10b shows that the number of Widmannstetter austenite in grains increased significantly compared to that without the magnetic field. This effect was ascribed to the increased energy fluctuation and structural fluctuation inside the molten pool due to the alternating magnetic field, which is conducive to the precipitation of a large number of small IGAs in a non-diffusive transition; also, the alternating magnetic field changed the heat distribution inside the molten pool. Therefore, the Widmannstetter austenite had sufficient time to nucleate and grow rapidly in ferrite grains in the form of side lath from grain boundaries. 

## 4. Conclusions

This study mainly introduced the influence of an alternating magnetic field on the distribution of elements and microstructure of welded joints and analyzed and characterized the texture distribution of welds, grain boundary orientation difference, and grain size. The following conclusions were obtained:
(1)Mo improved the pitting resistance of 2205 duplex stainless steel and promoted the precipitation of the σ  phase. Thus, Mo can maintain the uniformity of the two phases when matched with the content of Ni and N;(2)SEM and EDS analysis showed that the application of a 20 mT alternating magnetic field improved the uniform distribution of alloy elements in the two phases, promoted the diffusion and dissolution of Ni and N elements to γ phase, facilitated the transition from the α to γ phase, and reduced the precipitation of Cr_2_N;(3)The EBSD results of WM showed that the austenite content in WM increased by 6.5% after applying a 30 mT alternating magnetic field, further precipitating WA. Then, the WM texture strength was increased, and the maximum orientation densities of α and γ phases were increased by 5.12 and 0.89 compared to no magnetic field. The ferrite texture distribution shifted to the α orientation line and a strong {111} <110> texture appeared. The austenite texture was concentrated under the action of the alternating magnetic field, and the Cubic texture, copper texture, and goss texture were found under the alternating magnetic field of 30 mT;(4)The effect of an alternating magnetic field promoted the transformation of austenite Σ3 twin boundary to HABs, and the dynamic recrystallization of the ferrite phase reduced LABs. In addition, the average grain size of the α-phase and γ-phase under a 30 mT alternating magnetic field was reduced to 5.58 μm and 6.2 μm, respectively. 

## Figures and Tables

**Figure 1 materials-15-08741-f001:**
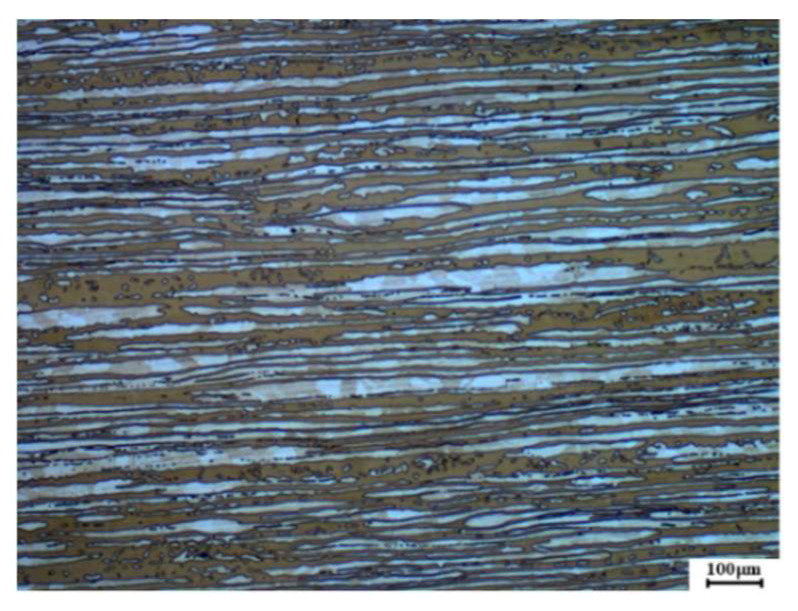
The microstructure of a 2205 duplex stainless steel plate.

**Figure 2 materials-15-08741-f002:**
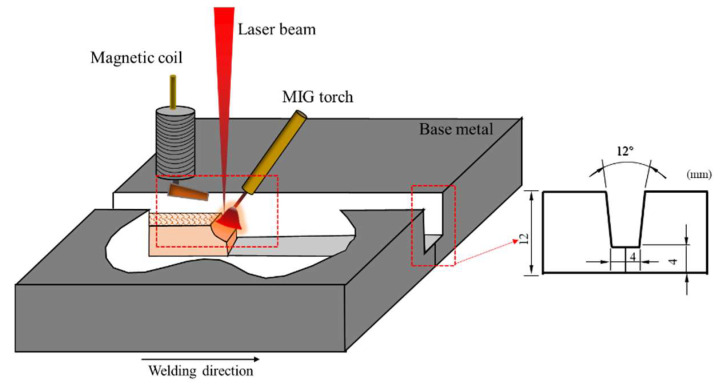
Schematic diagram of the experimental process and groove size.

**Figure 3 materials-15-08741-f003:**
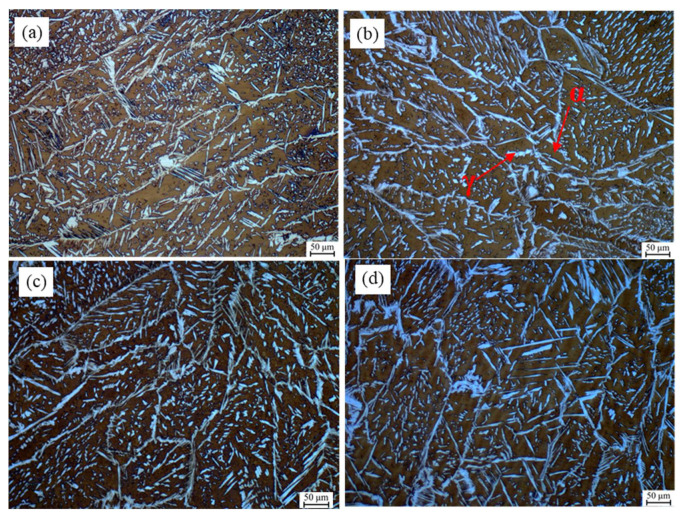
Microstructure of the filling weld under different alternating magnetic field parameters. (**a**) Without magnetic field; (**b**) f = 50 Hz B = 10 mT; (**c**) f = 50 Hz B = 20 mT; (**d**) f = 50 Hz B = 30 mT.

**Figure 4 materials-15-08741-f004:**
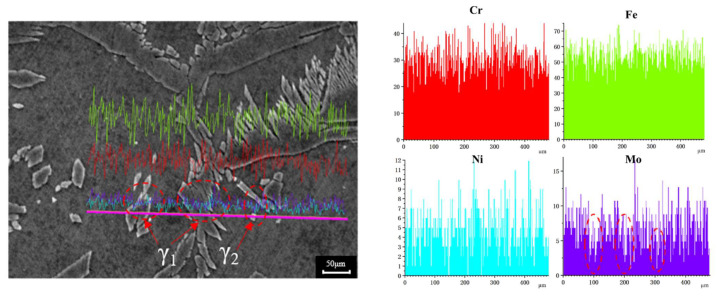
Line mapping analysis on the weld structure (color lines represent different elements).

**Figure 5 materials-15-08741-f005:**
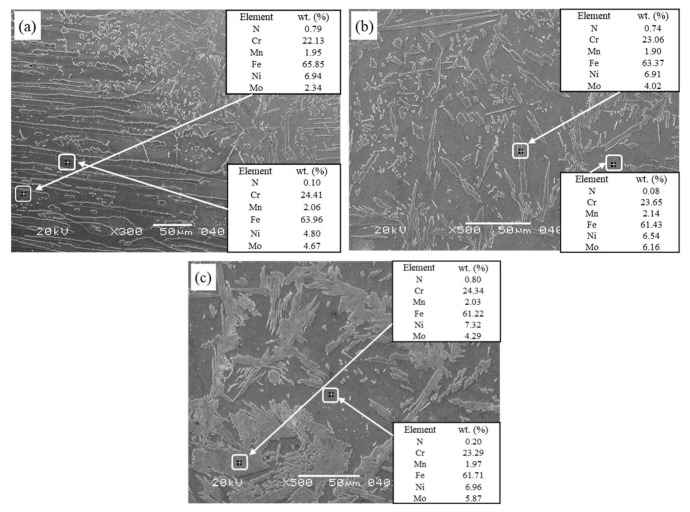
SEM and EDS analysis resulted in different welded joints. (**a**) BM and HAZ; (**b**) WM 0 mT; (**c**) WM 20 mT.

**Figure 6 materials-15-08741-f006:**
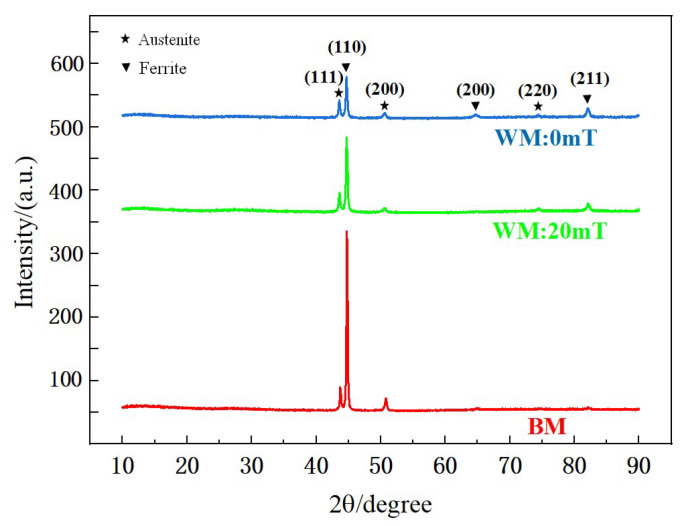
The XRD patterns in the different samples.

**Figure 7 materials-15-08741-f007:**
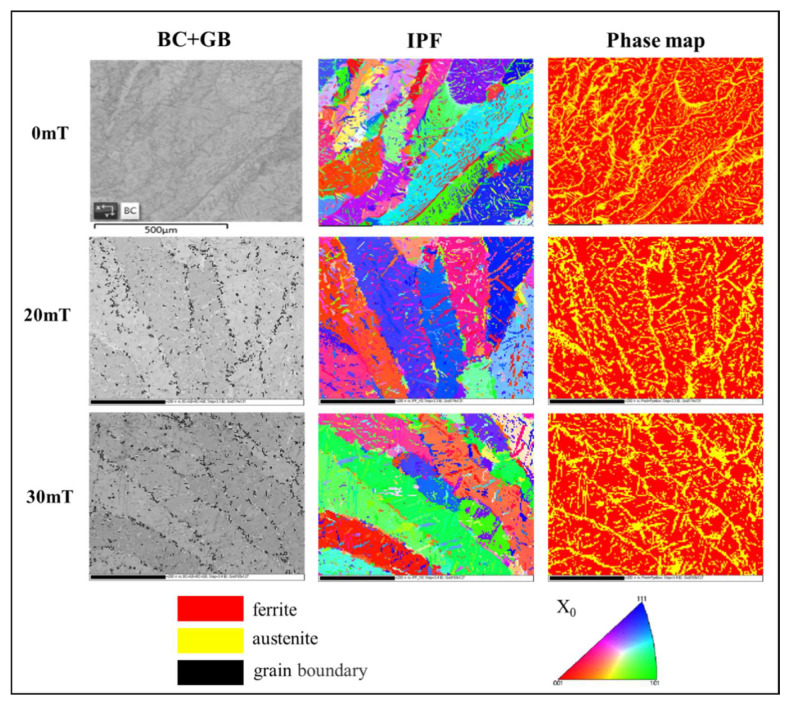
EBSD phase maps, IPFs in the RD direction, and band contrast of the weld under different alternating magnetic field intensities (the color codes are described in the image).

**Figure 8 materials-15-08741-f008:**
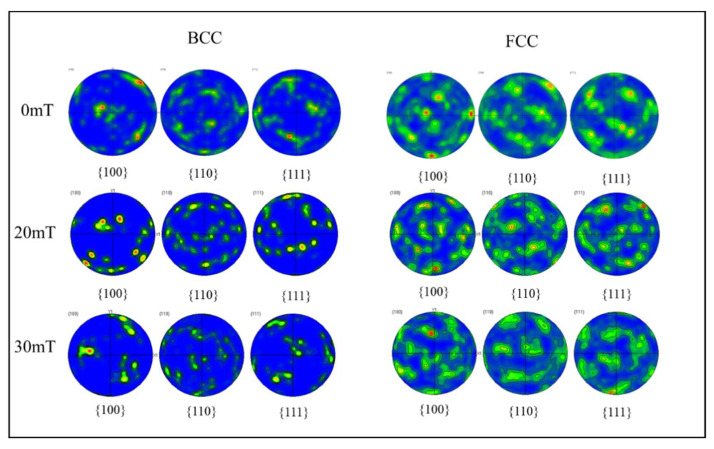
Pole figures of ferrite and austenite of weld under different alternating magnetic field intensities.

**Figure 9 materials-15-08741-f009:**
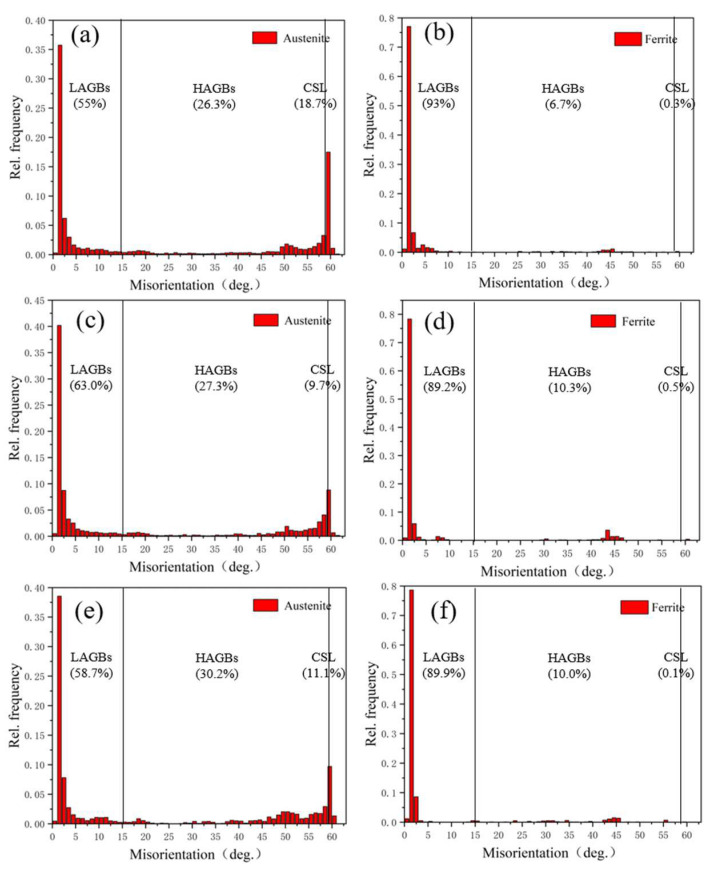
Misorientation angle distribution of ferrite and austenite grain boundaries of weld under different alternating magnetic field intensities. (**a**,**b**) 0 mT; (**c**,**d**) 20 mT; (**e**,**f**) 30 mT.

**Figure 10 materials-15-08741-f010:**
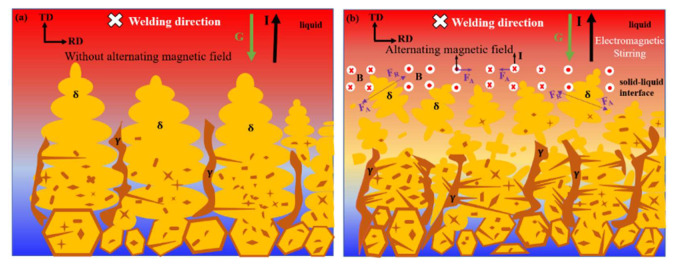
Effect of an alternating magnetic field on the dendrite arrangement during solidification. (**a**) Without alternating magnetic field; (**b**) Alternating magnetic field.

**Table 1 materials-15-08741-t001:** Main chemical composition of DSS2205 and ER2209 (wt.%).

Material	C	P	S	Cr	Ni	N	Mo	Mn	Si	Fe
2205 ER2209	0.024 0.019	0.023 0.016	0.001 0.0008	22.39 22.59	5.68 9.41	0.17 0.16	3.13 3.10	1.38 1.66	0.39 0.17	balance balance

**Table 2 materials-15-08741-t002:** Austenite and ferrite contents in WM under an alternating magnetic field.

The Alternating Magnetic Field Parameters	No Magnetic Field	f = 50 Hz B = 10 mT	f = 50 Hz B = 20 mT	f = 50 Hz B = 30 mT
γ (%)	39.1	40.7	43.4	44.6
α (%)	60.9	59.3	56.6	55.4

**Table 3 materials-15-08741-t003:** Mass percentage (%) of each element in austenite and ferrite in different areas of welded joints.

Position	Phase	Cr	Ni	Mo	N
BM	α	24.41	4.80	4.67	0.10
	γ	22.13	6.94	2.34	0.79
WM (0 mT)	α	23.65	6.54	6.16	0.08
	γ	23.06	6.91	4.02	0.74
WM (20 mT)	α	23.29	6.96	5.87	0.20
	γ	24.34	7.32	4.29	0.80

## Data Availability

The data used to support the findings of this study are available from the corresponding author upon request.

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
