# Peer review of "Microstructure and Texture Characterization of Duplex Stainless Steel Joints Welded by Alternating Magnetic Field-Assisted Hybrid Laser-GMAW Welding"

_materials, 2022, doi:10.3390/ma15248741_

Round 1
Reviewer 1 Report
The work is about the effect of alternating magnetic fields in the hybrid welding of duplex steel. This is an important research topic for further application in practice, but in general, the research carried out is not in sufficient depth. It is suggested that the entire text is thoroughly revised, discussion of the results is added and the manuscript is resubmitted. The following are the main remarks.
On all figures with scale markers, it is necessary to make them readable. Right now they are very small.
1. Section 2.1. The welding procedure is not adequately described. Firstly, Figure 2 shows a plate with a groove, i.e. not welding two plates, explain this point in the methodical part. Specify the wavelength of the laser beam for welding.
2. Section 2.2. Specify the radiation used for XRD analysis. Specify the scanning step for EBSD analysis.
3. Not specified method by which "No magnetic field" was determined? Add the information in section 2.
4. Numerical values of chemical elements content should be given according to the results of EDS mapping (Fig.4).
5. Figure 5. Use only the Latin alphabet, increase the font of the coordinate axes.
6. "The results show that Mo is helpful to improve the pitting resistance...". Refer to these results.
7. "SEM&EDS" should be replaced by "SEM and EDS".
8. The "&" symbol should be avoided throughout the text...
9. The terminology in the title of Table 3 is not clear, the title should be reworded. Specify the units of measurement in Table 3 (wt.% ?). Also, Table 3 duplicates the point analysis in Figure 6.
10. Section 3.1.3 General words are not appropriate in this section. A decrease in intensities of reflections (111)γ and (110)α is indicated, in this case numerical data of integral intensities of reflections should be given, then their ratios should be calculated and conclusions about the change of texture or volume fractions of the phases present should be drawn. "Most of the γ and α phases precipitate along the densely packed surfaces with the lowest surface energies: (111)γ and (110)α." Please revise the sentence. The use of the term "surface" is not clear in this section, replace it. "Ferrite phase is also formed in the (200)α and (211)α surfaces of the weld..." - is not a clear sentence.
Figure 7: Use only Latin letters.
"In addition, compared with the BM and WM without magnetic field, the peak width of each diffraction peak of 20mT alternating magnetic field has a certain change, indicating that the texture type and distribution and grain size of the WM are transformed after applying alternating magnetic field." It is clear from the text that the authors mean "full width at half maximum (FWHM)", but numerical data is also missing, we need to show numerically how FWHM changes depending on welding conditions. Also a link to literature describing the effect of texture type and distribution on FWHM should be added.
11. The poor quality of figure 8. The whole section 3.2.1 needs a thorough revision.
Reviewer 2 Report
1. The aim of the study should be more clearly emphasised. The Authors should justify why their work is novel in comparison to available literature.
2. Figure 3 - Authors should mark with arrows the microstructure components, so it will be easier for Readers to understand where are particular phases.
3. How did Authors calculate the content of austenite and ferrite?
4. The EDS mappings in Figure 4 shows almost nothing. Their quality should be improved, maybe longer time of acquisition is needed.
5. Figure 5 - maybe instead of a linear measurement a proper EDS maps would be better to show differences in a chemical composition.
6. Figure 7 - please remove non-english words.
7. What was the accuracy of the EBSD measurements? What was the cleaning procedure?
8. Figure 9 should be improved. The Authors put pole figures together with the legends with the intensity, however, it is so small that it is not possible to read it. Therefore, it should be enlarged or erased from the image.
9. ODF maps in Figure 11 must be enlarged. They are too small and it is difficult to read them.
Round 2
Reviewer 1 Report
Thank you for the responses to the remarks provided. This paper can now be published.
Reviewer 2 Report
Thank you for the answers.